# Physical Activity, Physical Fitness and Energy Intake Predict All-Cause Mortality and Age at Death in Extinct Cohorts of Middle-Aged Men Followed-Up for 61 Years

**DOI:** 10.3390/jcm14072178

**Published:** 2025-03-22

**Authors:** Alessandro Menotti, Paolo Emilio Puddu

**Affiliations:** 1Association for Cardiac Research, 00182 Rome, Italy; amenotti2@gmail.com; 2EA 4650, Signalisation, Électrophysiologie et Imagerie des Lésions D’ischémie Reperfusion Myocardique, Normandie Université, UNICAEN, 14000 Caen, France

**Keywords:** physical activity, physical fitness, energy intake, all-cause mortality

## Abstract

**Objective**: Working physical activity, physical fitness and energy intake were studied for their effect on all-cause mortality and age at death in residential cohorts followed-up for 61 years. **Material and Methods**: There were two residential cohorts of middle-aged men examined in 1960 with a total of 1712 subjects, and three indexes were measured, i.e., physical activity by a questionnaire (three classes—sedentary, moderate, vigorous: Phyac), physical fitness, estimated by combining arm circumference, heart rate, and vital capacity by Principal Component Analysis, whose score was divided into three tertile classes (low, intermediate, and high: Fitscore), and energy intake in Kcalories, estimated by dietary history divided into three tertile classes (low, intermediate, high: Calories), plus five traditional cardiovascular risk factors (age, cigarette smoking, body mass index, systolic blood pressure, and serum cholesterol). Cox models were used to predict all-cause mortality as a function of those adjusted indexes. Multiple linear regression models were used to predict age at death as a function of the same co-variates and a larger number of them. At the 61-year follow-up, 1708 men had died. **Results**: There were large correlations across the three indexes. Prediction of all-cause mortality showed the independent and complementary roles of the three indexes to all be statistically significant and all protective for their highest levels. However, the Fitness score outperformed the role of Phyac, while the role of Calories was unexpectedly strong. The same outcome was found when predicting age at death, even in the presence of 25 covariates representing risk factors, personal characteristics, and prevalent major diseases. **Conclusions**: Working physical activity, a score of physical fitness and energy intake, seems directly related to lower all-cause mortality and to higher age at death, thus suggesting a large part of independence.

## 1. Introduction

Old [1] and relatively recent [2] reviews have clearly shown evidence of the beneficial role of physical activity on health status and the danger of sedentary habits. Similar findings were provided in the Italian Rural Areas of the Seven Countries Study of Cardiovascular Diseases (SCS), which started in 1960, where participants were classified for their physical activity into three classes, i.e., sedentary, moderate, and vigorous, and the end-point was made to be all-cause mortality, several single causes of death, and age at death during extremely long follow-up periods [3]. The classification of physical activity was based only on apparent engagement at work, but, despite its rough characteristics, it performed in a rather good way. Previously, in 1992, we carried out an analysis where, together with the basic physical activity classification, some alleged fitness components were considered, showing good relations with 25-year all-cause mortality [4].

More recently, a research group from the SCS published two contributions comparing the role of physical activity versus that of a Fitness score derived from the combined levels of arm circumference, heart rate, and vital capacity in predicting major lethal events in a group of European cohorts of middle-aged men [5,6]. The present analysis aims to replicate that analysis in the Italian areas of the SCS with the addition of another variable, i.e., energy intake, whose connection with energy expenditure may help to validate the previous approach.

Physical activity and physical fitness are defined in different ways, and following some classical proposals, physical activity can be defined as “*any bodily movement produced by skeletal muscles that results in energy expenditure*” [1], while physical fitness is defined as “*the ability to carry out daily tasks with vigor and alertness, without undue fatigue and with ample energy to enjoy leisure-time pursuits and to meet unforeseen emergencies*” [1]. Sometimes, confusion has arisen considering these two characteristics, which clearly have some connections. However, physical fitness can be partly due to genetic traits and improved or promoted by physical activity.

## 2. Material and Methods

**Population and measurements:** The analysis was run on data from Italian Rural Areas (IRAs) of the Seven Countries Study of Cardiovascular Diseases conducted with 1712 men aged 40–59 years at the time of entry examination in 1960, representing 98.7% of the participants invited. Details of these residential cohorts may be found elsewhere [3].

The variables used for analysis were (A) working physical activity classification (Phyac) derived from the type of work with the addition of a few non-standardized questions, classified as low, intermediate, and high (roughly corresponding to sedentary, moderate, and vigorous); leisure physical activity was not considered since it was practically absent among males in rural communities in the early 1960s; (B) indicators of physical fitness (Fitscore), namely (i) arm circumference (in mm), following the technique reported in the WHO Manual [7], with the crude measurement cleaned for the contribution of subcutaneous tissue using a formula that included the tricipital skinfold thickness [8]; this characteristic was considered an indicator of muscular mass; (ii) heart rate (beats/min) derived from a resting ECG; this characteristic was considered an indicator of cardio-circulatory fitness; (iii) vital capacity (L/m^2^), following the technique reported in the WHO Manual [7] and using the best of 2 attempts for analysis; this characteristic was considered a respiratory indicator of fitness; (C) energy intake calculated in daily Kilocalorie intake, derived from a dietary survey based on dietary history, using a questionnaire administered by trained, experienced, and supervised technicians and computed from local food tables [9]; and (D) other variables used as possible confounders in multivariate predictive analysis, which were (1) age (years), approximated to the nearest birthday; (2) cigarettes smoked on average per day (N/day), adopted since preliminary analyses showed that, in the long run, ex-smokers had a risk rather similar to non-smokers (both groups being classified with zero cigarettes); (3) body mass index (kg/m^2^), following the technique reported in the WHO Manual [7]; (4) systolic blood pressure (mmHg), measured in supine position at the end of a physical examination using a mercury sphygmomanometer, following the technique reported in the WHO Manual [7], adopting the average of two measurements taken 1 min apart as an analytical variable; (5) serum cholesterol (mmol/L) measured in casual blood samples following the technique by Anderson and Keys [10]. Many more risk factors and personal characteristics were used as possible confounders in a final multiple linear regression with age at death as a dependent variable.

**Mortality data:** During the 61 years, out of 1712 men enrolled at baseline, there were 1708 deaths, 3 men were still alive, with an age ranging from 102 to 106 years, and 1 man was lost to follow-up after year 50 of follow-up when he was aged 91 years. End-points for this analysis were all-cause mortality and, for those who died, age at death in years. However, the 4 men still alive or lost to follow-up received an estimate age at death by adopting the age at which they were last seen alive, and then they were included in all the analyses.

Age at death is an old demographic metric that has been recently re-evaluated [11]. However, its use in population cohort studies is legitimate only if the cohorts are extinct or nearly extinct.

**Statistical Analysis:** Phyac was used as originally defined and classified in three classes (low, intermediate, high), roughly corresponding to sedentary, moderate, and vigorous physical activity (Phyac1, Phyac2, Phyc3). Fitscore was a factor score derived from a Principal Component Analysis (PCA), where the three indexes of fitness were used for the computation. The PCA coefficients of the three indexes were −0.1404 for heart rate, +0.6812 for vital capacity, and +0.6433 for arm circumference (see Appendix A). Each individual had a value of Fitscore represented by the factor score of the PCA, and the rank list was divided into 3 tertile classes, corresponding to low, intermediate, and high levels (Fitscore1, Fitscore2, Fitscore3). Calories were treated, in the majority of analyses, into three tertile classes (low, intermediate, high) similar to Fitscore. The three classes of Fitscore and Calories had different numerical sizes compared to Phyac as a consequence of the different procedure adopted for their creation.

The mean baseline levels of the various variables were computed. The mean levels of arm circumference, heart rate, and vital capacity were distributed into three classes of Phyac, Fitscore, and Calories, and ANOVA was computed across the three levels.

Tests of the predictive power of the three indexes were carried out as follows: (a) Kaplan–Meier survival versus all-cause mortality separately for Phyac, Fitscore, and Calories, each divided into three classes; (b) Cox proportional hazard models with all-cause mortality as the end-point run in three different modes, i.e., with Phyac alone (model 1), then with Phyac and Fitscore (model 2), and then with Phyac, Fitscore, and Calories (model 3) as covariates, with the addition of, as possible confounding variables, age, cigarette smoking, body mass index, systolic blood pressure, and serum cholesterol. The three main covariates were used as divided into three classes (the lowest being used as a reference). The Cox model with all-cause mortality could be run, despite the practical extinction of the cohort, since it included the role of time and survival; (c) multiple linear regression (MLR) models with age at death as the end-point were run in three different modes, with Phyac alone (model 1), then with Phyac and Fitscore (model 2), and then with Phyac, Fitscore, and Calories (model 3) as covariates with the addition of, as confounding variables, age, cigarette smoking, body mass index, systolic blood pressure, and serum cholesterol. The three main covariates were again used as divided into three classes (the lowest being used as a reference). For both Cox and MLR models, T tests comparing coefficients were computed.

Finally, another MLR model was computed using age at death as the end-point, indicators of physical activity, Fitscore, and Calories, and a long list of other risk factors, personal characteristics, and major prevalent diseases as possible confounders. On this occasion, several variables had to be excluded due to either being part of the three indicators or showing multicollinearity problems with the same variables. These were arm circumference, heart rate, vital capacity, forced expiratory volume, tricipital skinfold, and a dietary score divided into three classes.

## 3. Results

**Baseline variables:** The baseline mean levels of the variables used in the analysis are given as a reference in Table 1. They reflect the levels common among men in rural environments in Italy in the 1960s, with relatively high mean levels of blood pressure and smoking habits and relatively low mean levels of serum cholesterol. Also, energy intake was relatively high but justified by the high physical activity levels. Phyac, divided into three classes, showed an excess of vigorous physical activity justified by the large number of farmers in the population sample.

**Phyac and Fitscore versus Caloric intake:** In Table 2, the mean levels of Calorie intake regularly increase from Class 1 to Class 3 of the two indexes, and in both cases, the ANOVA across the three classes is highly significant for heterogeneity and trend.

**Phyac and Fitscore versus indicators of fitness:** In Table 3, there are increasing levels of arm circumference and vital capacity across the three classes of Phyac, Fitscore, and Calorie indexes, while the reverse is the case for heart rate. In all cases, the ANOVA is highly significant for heterogeneity and trend. Using the original units of measurement, the correlation coefficient (R) of Fitscore versus Calories was 0.25 and that of Fitscore versus Phyac was 0.24, whereas that of Phyac versus Calories was 0.21, with all of these being highly significant.

**Prediction of 61-year mortality and age at death by Phyac and Fitscore:** Kaplan–Meier survival curves with ±95% confidence intervals (Figure 1) for Phyac showed larger confidence intervals in Class 1 with few men as compared to Class 3 and an acceptable separation among the three classes. However, Classes 2 and 3 largely overlapped. The three survival curves for Fitscore with ±95% confidence intervals (Figure 2) were instead largely segregated, with longer survival for intermediate and high levels. Also, survival curves with ±95% confidence intervals for Calories (Figure 3) had an overlap between Classes 2 and 3, but they were rather separated with shorter survival for the low class. It is noteworthy that for Fitscore and Calories, the three classes had the same number of individuals. All characteristics showed strong significance levels for *p* of log rank chi^2^ (respectively, *p* < 0.0017 for Phyac, *p* < 0.0001 for Fitscore, and *p* < 0.0001 for Calories).

Cox proportional hazard models predicting all-cause mortality in 61 years showed a negative algebraic sign for all coefficients, and all had significant *p* values, except marginally for Phyac2 in models 2 and 3 (Table 4). Fitscore coefficients were all significant and, in general, greater than those of Phyac, although the differences never reached a significant level. Also, the coefficients for Calories in model 3 were statistically significant. In all cases, low levels of Phyac, Fitscore, and Calories had adverse effects, while the opposite was the case for high levels.

Similar findings were seen for MLR models predicting age at death in 61 years of follow-up (Table 5). The coefficients of Phyac, Fitscore and Calories (which should be interpreted as relative risk versus the reference variable) were all positive and significant, with beneficial effects for their high levels. The addition of Fitscore in model 2 and that of Calories in model 3 were associated with slight non-significant decreases in the levels of Phyac coefficients. The advantages for age at death associated with high levels of the three major covariates ranged from 1.9 to 3.5 years.

A series of 24 comparisons between couples of major determinants’ coefficients were carried out as follows: Phyac versus Fitscore in the same models; Phyac versus Phyac in different (parallel) models; and Fitscore versus Fitscore in different (parallel) models. None of the comparisons, conducted independently for Cox and MLR models, were statistically significant.

The MLR model is reported in Table 6 with age at death as the end-point and 35 possible confounders (including 4 reference variables). Among them, there were social data, family health problems, behavior lifestyle, anthropometric measurements, clinical signs, and prevalent diseases. The findings confirmed the strong predictive power of Phyac, Fitscore, and Calories, even in the presence of so many other potential predictors that, in part, were statistically significant. It is hard to describe the different predictive roles of the three indexes, but this MLR model could be the right choice for trying to classify their roles since estimates are adjusted for many possible confounders. The worst is Phyac since only a high level physical activity carries a significant coefficient (not the coefficient of moderate physical activity). All coefficients for Fitscore and Calories are significant, but the hazard ratios for Fitscore are definitely larger than those of Calories, suggesting that Fitscore is the strongest index among the three tested in this model. This multivariate approach confirms the univariate impression from the Kaplan–Meier curves (±95% confidence intervals), whereby a clearcut separation was seen for Fitscore curves (Figure 2), whereas the curves more or less overlapped for Phyac (Figure 1, Classes 2 and 3 versus 1) and Calories (Figure 3, Classes 2 and 3 versus 1). This implies that the best univariate result was that of Fitscore versus Phyac or Calories.

## 4. Discussion

The purpose of this analysis was to validate a rough physical activity classification using characteristics bound to physical fitness and caloric intake. Actually, we had another source of validation, consisting of caloric expenditure estimated by ergonometric measurements performed in men, classified in the same way, with mean levels around 2400 Kcal for sedentary activity, less than 3000 for moderate activity, and >3000 for vigorous activity [12]. Unfortunately, the original individual data are not available anymore and could not be compared with this analysis.

The present findings suggest that a score of physical fitness (Fitscore), produced by combining the role of arm circumference, heart rate, and vital capacity, is equally or even more predictive of 61-year mortality deaths and age at death when compared with the original classification of Phyac. Moreover, even the three classes of energy intake in Kcal are well related to the fitness indexes and also predictive of events. An interesting fact is that, despite the good relationship across Phyac, Fitscore, and Calories, these three characteristics produced independent, significant, and additive coefficients when forced into the same multivariate models. Proper tests were conducted, and no problems were found in terms of multicollinearity, confirmed by high levels of tolerance.

The apparent conclusion is that the three classes of physical activity in the original classification are well related to the indicators of fitness, involving muscular mass (arm circumference) and circulatory (heart rate) and respiratory (vital capacity) functions, and to energy expenditure, justifying their valuable predictive power of events. Moreover, the strong connections between Fitscore and Phyac are confirmed by the fact that 80% of men in the high class for Fitscore were classified in a high class for Phyac.

A somewhat unexpected but interesting finding consisted of the favorable relationship of energy intake with mortality and age at death. This can be explained by the fact that, in this rural population, men with higher energy intake were mainly the same men who burnt many Calories during long working days in the fields. In fact, from a metabolic point of view, energy intake, in the case of probable energy balance, is a good estimate of energy expenditure. The final MLR model (Table 6) confirmed the large independence of the three indexes from the possible confounding role of many other covariates for their role in predicting age at death.

A relatively simple score, like that suggested here, indicated that arm circumference, heart rate, and vital capacity are objective indicators of muscular, cardio-circulatory, and respiratory fitness and contribute significantly to the current knowledge in this field.

The three classes of occupation physical activity in the original Seven Countries Study classification [3] are well related to indicators of fitness, including muscular mass (arm circumference) and circulatory (heart rate) and respiratory (vital capacity) functions, thus leading to their valuable predictive power of events. The Fitscore derived from the above indicators represents an outperforming and powerful predictor of all-cause death and age at death [5]. The literature does not offer contributions that are similar to the present one due to different definitions of physical activity and physical fitness and the rare use of energy intake, so comparisons with our findings can only be indirect.

Despite the satisfactory findings of this analysis, we acknowledge the existence of various limitations in our approach. Namely, the study was restricted to men since women were not considered in the roster, the age range of the participants was limited to 40 to 59 years, and the overall sample was relatively small, although this was partly counterbalanced by the extremely long follow-up, reaching the death of the whole cohort. The technical procedure for the classification of physical activity was extremely rough, but in those times (i.e., 65 years ago), valuable tools for this purpose were not available, although even more recently, there have been studies using a classification of physical activity based on a single question or self-reported information [1,2]. The absence of substantial leisure physical activity was a fact and not a forgotten issue. All the above limitations seem to contradict the strong relationships found between our physical activity classification and the levels of Fitscore and energy intake, as well as with several end-points, as previously published [4,5].

The dietary history used in the dietary survey might be considered not reliable, but at those times, it was among the best available tools. In this case, we should forget a large number of dietary studies conducted over the last century, beyond considering that many recent investigations are based on questionnaires that look for the frequency of consumption of certain food groups. Another field examination was held at around mid-way in the follow-up (in year 31), but only a quarter of the original population was still alive (aged 71 to 90 years), and some critical variables for this analysis were not measured again, thus preventing us from studying the effect of risk factor changes [3]. On the other hand, we rely on the long-term predictive power of at least some risk factors since this was shown for durations of up to 40 years from the time of measurement in the same study population and reported elsewhere [3]. At the 31-year point, re-examination data on frailty and similar problems among the elderly were not collected by our research group since our main interest was focused on characteristics allowing for a long life expectancy and not on describing the characteristics of those who reached old age when this goal was obtained.

We used a single question or few questions to define the levels of working physical activity, and a similar approach was followed by others, but the conclusion was that it was not enough to classify in a proper way people with sedentary habits [13]. Other investigators measured the time spent doing physical activity to reach a valuable classification [14]. There were several other methods used to classify physical activity, including self-reported information [15], the use of activity pattern questionnaires [16], and the estimate of metabolic equivalents [17].

In most studies, physical fitness was defined by the outcome of maximal exercise testing, either comparing physical activity with physical fitness or considering only physical fitness [15,16,17,18,19,20,21,22]. In general, comparative studies showed a better performance in physical fitness than physical activity, as partly was the case in our previous experience [5,6]. Among studies focused on single fitness indicators, two contributions deserve mention since they stress the beneficial role of physical activity on respiratory function and the role of the latter as a predictor of all-cause mortality [23,24]. Moreover, it is interesting to learn that cardiometabolic risk factors can be improved by increasing muscular strength [25].

There are recent contributions, like a large metanalysis, suggesting that high levels of working physical activity are not protective against cardiovascular diseases and all-cause mortality [26]. This conclusion seems to include both recent studies and older studies published before 1989. On the other hand, the same meta-analysis suggests the protective role of leisure physical activity. From this point of view, our population did not engage in leisure physical activity that would have been difficult to describe in any way because, in those times, complex questionnaires on leisure physical activity were not available. Moreover, nowadays, the levels of working physical activity are probably smaller than those observed in a rural environment in the 1960s.

In conclusion, our findings point to the protective role of vigorous physical activity during work against all-cause mortality and age at death based on variables that were at least roughly measurable in the middle of the last century, together with a simple score of fitness and energy intake, whose roles are related but also cumulative. The present contribution also underscores the role of Calories, which should be used as an index of energy expenditure and balance when integrated with Fitscore and Phyac. Calories should then be measured in all individuals as they contribute, at a high level, to prolonging life. Despite the methodological limitations and the uncertainty of the conclusions, the combination of three simple risk factors in the so-called Fitscore seems to be an innovative proposal that probably deserves more testing and clear comparison with the outcome of exercise testing as the most common procedure to estimate physical fitness [5].

We recommend that further comparative studies be carried out, including the female sex and considering leisure physical activity, which nowadays is a prevalent risk factor for most people. On the other hand, from a technical point of view, the measurements of Phyac, Fitscore, and Calories are no longer as complex as in the past, mainly due to the availability of modern spirometers, which are definitely superior to the old-fashioned ones used in this study. The questionnaires have also been updated, and these should be adopted in future investigations.

## Figures and Tables

**Figure 1 jcm-14-02178-f001:**
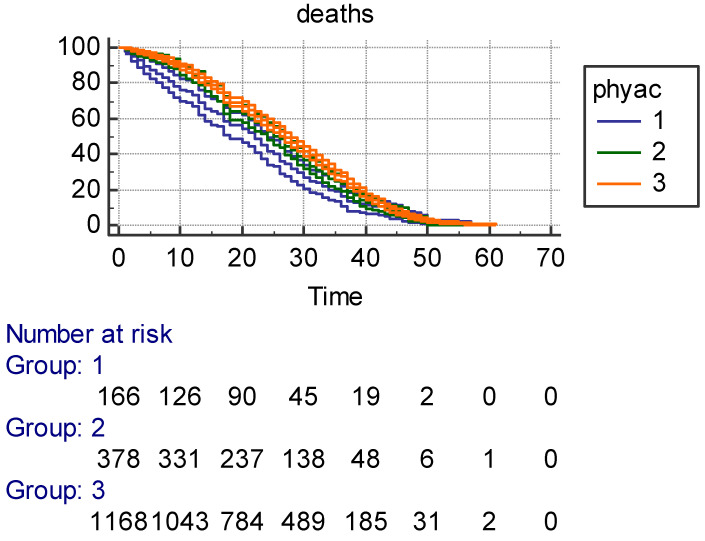
Kaplan–Meier survival in 61 years (±95% confidence intervals) as a function of 3 classes of Phyac (in box)—Group 1 = low; Group 2 = intermediate; Group 3 = high—with individuals at risk at different time points.

**Figure 2 jcm-14-02178-f002:**
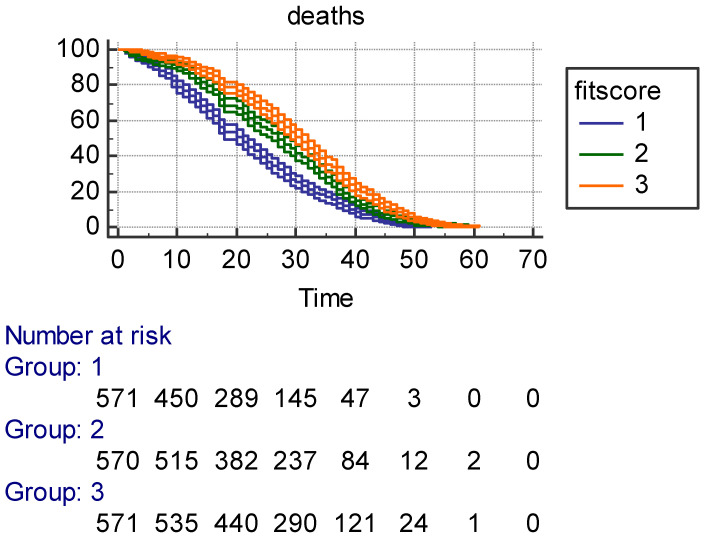
Kaplan–Meier survival in 61 years (±95% confidence intervals) as a function of 3 tertile classes of Fitscore (in box)—Group 1 = low; Group 2 = intermediate; Group 3 = high—with individuals at risk at different time points.

**Figure 3 jcm-14-02178-f003:**
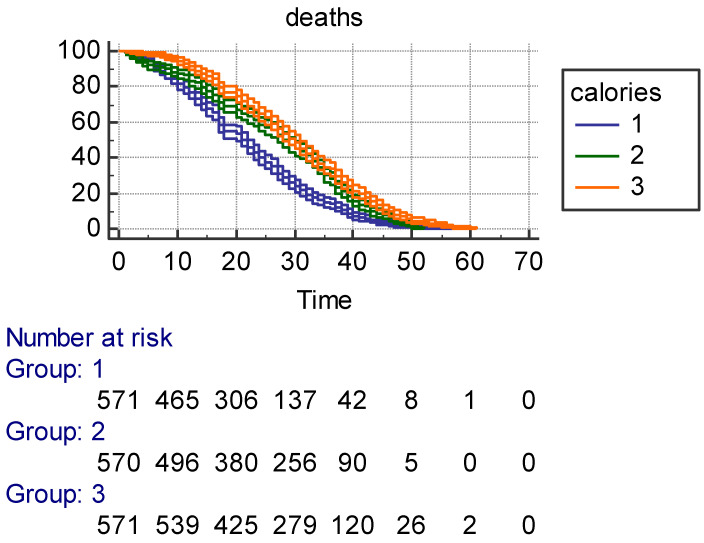
Kaplan–Meier survival in 61 years (±95% confidence intervals) as a function of 3 tertile classes of Calories (in box)—Group 1 = low; Group 2 = intermediate; Group 3 = high—with individuals at risk at different time points.

**Table 1 jcm-14-02178-t001:** Baseline mean levels of variables used in the analysis.

Variable		
**Physical activity class**	**N**	**% (SE)**
Low	166	9.7 (0.7)
Intermediate	378	22.1 (1.0)
High	1168	68.2 (1.1)
**Fitness variables**	**Mean**	**SD**
Arm circumference, mm	268.6	23.6
Heart rate, beats/min	71.3	12.9
Vital capacity, L/m^2^	1.65	0.24
**Calories**		
Daily intake	3112	647
Calories, tertile 1	2463	346
Calories, tertile 2	3108	131
Calories, tertile 3	3766	517
**Confounding variables**		
Age, years	49.1	5.1
Cigarette, N/day	8.7	9.5
Body mass index, kg/m^2^	25.2	3.7
Systolic blood pressure, mmHg	143.6	21.0
Serum cholesterol, mmol/L	5.21	1.06

N = number; SE = standard error; SD = standard deviation.

**Table 2 jcm-14-02178-t002:** Mean values of caloric intake in 3 classes of Phyac and Fitscore.

Variable	Mean (SD)	Mean (SD)
Class	**Phyac low**	**Fitscore low**
N	166	571
Energy, Kcal/day	2816 (618)	2919 (614)
Class	**Phyac intermediate**	**Fitscore intermediate**
N	378	570
Energy, Kcal/day	2962 (602)	3164 (650)
Class	**Phyac high**	**Fitscore high**
N	1168	571
Energy, Kcal/day	3203 (643)	3254 (629)
ANOVA across classes	*p* < 0.0001	*p* < 0.0001

N: number; SE: standard error; SD: standard deviation.

**Table 3 jcm-14-02178-t003:** Mean values of indicators of fitness in 3 classes of Phyac, Fitscore, and Calories.

Variable	Mean (SD)	Mean (SD)	Mean (SD)
	**Phyac low** N = 166	**Fitscore low** N = 571	**Calories low** N = 571
Arm circumference	259.4 (5.2)	255.6 (23.3)	264.8 (25.5)
Heart rate	77.4 (14.8)	81.3 (13.4)	73.8 (14.0)
Vital capacity	1.59 (0.27)	1.45 (0.21)	1.58 (0.25)
	**Phyac intermediate** N = 378	**Fitscore** **intermediate** N = 570	**Calories intermediate** N = 570
Arm circumference	268.1 (25.6)	268.0 (19.9)	268.6 (21.6)
Heart rate	74.3 (13.5)	69.2 (9.2)	70.8 (12.1)
Vital capacity	1.61 (0.25)	1.65 (0.15)	1.66 (0.23)
	**Phyac high** N = 1168	**Fitscore high** N = 571	**Calories high** N = 570
Arm circumference	270.0 (22.1)	282.0 (19.7)	272.2 (22.9)
Heart rate	69.5 (11.9)	63.4 (8.2)	69.3 (12.1)
Vital capacity	1.67 (0.21)	1.84 (0.18)	1.70 (0.22)
ANOVA			
Arm circumference	<0.0001	<0.0001	<0.0001
Heart rate	<0.0001	<0.0001	<0.0001
Vital capacity	<0.0001	<0.0001	<0.0001

Units of measurement as in Table 1.

**Table 4 jcm-14-02178-t004:** Multivariate models predicting all-cause mortality (Cox) as a function of 3 classes of Phyac, Fitscore, and Calories, adjusted for 5 covariates.

	Coefficient	*p* Value	Hazard Ratio	95% CI	
**COX model (1) predicting all-cause mortality with Phyac only**					
Phyac 1	Reference	----	----	----	
Phyac 2	−0.1948	**0.0383**	0.82	0.68	0.99
Phyac 3	−0.2740	**0.0004**	0.76	0.64	0.90
**COX model (2) predicting all-cause mortality with Phyac and Fitscore**					
Phyac 1	Reference	-----	-----	-----	
Phyac 2	−0.1872	**0.0465**	0.83	0.69	1.00
Phyac 3	−0.2226	**0.0089**	0.80	0.68	0.95
Fitscore 1	Reference	-----	-----	-----	
Fitscore 2	−0.2286	**0.0002**	0.80	0.71	0.90
Fitscore 3	−0.2616	**0.0001**	0.77	0.68	0.87
**COX model (3) predicting all-cause mortality with Phyac, Fitscore, and Calories**					
Phyac 1	Reference	-----	-----	-----	
Phyac 2	−0.1731	0.0658	0.84	0.70	1.01
Phyac 3	−0.1830	**0.0342**	0.83	0.70	0.99
Fitscore 1	Reference	-----	-----	-----	
Fitscore 2	−0.2010	**0.0012**	0.82	0.72	0.92
Fitscore 3	−0.2428	**0.0002**	0.78	0.70	0.89
Calories 1	Reference	-----	-----	-----	
Calories 2	−0.2394	**0.0001**	0.79	0.70	0.89
Calories 3	−0.1998	**0.0023**	0.82	0.72	0.93

CIs: confidence intervals. In **bold** are significant *p* values.

**Table 5 jcm-14-02178-t005:** Multiple linear regression (MLR) models predicting age at death as a function of 3 classes of Phyac, Fitscore and Calories, adjusted for 5 covariates.

	Coefficient	*p* Value	95% CI	
**MLR Model (1) Predicting Age at Death with Phyac Only; R^2^ = 0.0964**				
Phyac 1	Reference	-----	-----	
Phyac 2	2.4898	**0.0154**	0.48	4.50
Phyac 3	3.2097	**0.0005**	1.41	5.00
**MLR model (2) predicting age at death with Phyac and Fitscore; R^2^ = 0.1093**				
Phyac 1	Reference	-----	-----	
Phyac 2	2.3688	**0.0202**	0.37	4.37
Phyac 3	2.5104	**0.0064**	0.71	4.31
Fitscore 1	Reference	**-----**	-----	
Fitscore 2	2.5132	**0.0002**	1.15	3.76
Fitscore 3	3.5287	**0.0001**	2.15	4.91
**MLR model (3) predicting age at death with Phyac, Fitscore, and Calories; R^2^ = 0.1165**				
Phyac 1	Reference	-----	-----	
Phyac 2	2.1578	**0.0339**	0.17	4.15
Phyac 3	1.9303	**0.0381**	0.11	3.75
Fitscore 1	Reference	-----	-----	
Fitscore 2	2.1554	**0.0012**	0.85	3.46
Fitscore 3	3.2317	**0.0001**	1.85	4.61
Calories 1	Reference	-----	-----	
Calories 2	2.2916	**0.0005**	0.99	3.59
Calories 3	2.4836	**0.0004**	1.10	3.87

CIs: confidence intervals. Note: in these MLR models with dichotomic variables, the levels of coefficients indicate relative risk versus the respective reference variables. In **bold** are significant *p* values.

**Table 6 jcm-14-02178-t006:** MLR model with age at death as end-point and 35 variables of different types as predictors, including Phyac, Fitscore, and Calories. R^2^ = 0.1532.

Variable	Coefficient	*p* Value	Delta	Effect	95% Cl	
Intercept	95.0153	<0.0001	----	----	----	
Age, years	0.1942	**0.0009**	5	0.97	0.40	1.54
High socio-economic status, 1-0	2.0930	**0.0284**	1	2.09	0.22	3.96
Father early death, 1-0	−1.4110	**0.0274**	1	−1.41	−2.66	−0.16
Moher early death, 1-0	−1.8418	**0.0045**	1	−1.84	−3.11	−0.57
Family CVD, 1-0	−0.2628	0.6250	1	−0.26	−1.32	0.79
Married, 1-0	1.7756	**0.0478**	1	1.78	0.02	3.53
Smoker, 1-0	reference	----	----	----	----	
Ex smoker, 1-0	1.3260	0.0955	1	1.33	−0.23	2.88
Never smoker, 1-0	2.9395	**<0.0001**	1	2.94	1.70	4.18
Body mass index, kg/m^2^	−0.4072	**0.0053**	3.7	−1.51	−2.57	−0.45
Trunk/height, ratio	−0.0640	0.7250	1.5	−0.10	−0.63	0.44
Shoulder pelvis shape, ratio	−7.4007	**0.0409**	0.1	−0.74	−1.45	−0.03
Laterality/linearity index, ratio	−0.2384	0.1501	1.8	−0.43	−1.01	0.15
Subscapular skinfold, mm	0.2544	**0.0009**	6	1.53	0.62	2.43
Systolic blood pressure, mmHg	−0.1113	**<0.0001**	20	−2.23	−2.78	−1.67
Serum cholesterol, mg/dl	−0.0232	**0.0006**	40	−0.93	−1.46	−0.40
Urine protein, 1-0	−1.4725	0.14091	1	−1.47	−3.43	0.49
Urine glucose, 1-0	0.0651	0.9906	1	0.07	−10.73	10.86
Corneal arcus, 1-0	−1.9679	**0.0110**	1	−1.97	−2.79	−0.36
Xanthelasma 1-0	−4.6000	**0.0323**	1	−4.60	−4.40	−0.20
Major CVD, 1-0	−3.3550	**0.0089**	1	−3.36	−5.87	−084
Cancer, 1-0	−20.0279	**<0.0001**	1	−20.03	−29.46	−10.60
Diabetes, 1-0	−2.2521	0.6760	1	−2.25	12.81	8.31
Chronic bronchitis, 1-0	−2.8434	**0.0088**	1	−2.84	−4.97	−0.72
Silent ECG abnormalities, 1-0 (*)	−1.1593	0.3772	1	−1.16	−3.79	1.47
Positive exercise ECG, 1-0 (*)	−1.7726	0.3161	1	−1.77	−5.24	1.69
Low physical activity, 1-0	reference	----	----	----	----	
Moderate physical activity, 1-0	1.5849	0.1180	1	1.58	−0.40	3.57
High physical activity	2.2821	**0.0202**	1	2.28	0.36	4.21
Low Fitscore, 1-0	reference	----	----	----	----	
Intermediate Fitscore, 1-0	2.3545	**0.0019**	1	2.35	0.87	3.84
High Fitscore, 1-0	3.3821	**0.0004**	1	3.38	1.53	5.23
Low Calories, 1-0	reference	----	----	----	----	
Intermediate Calories, 1-0	1.8547	**0.0047**	1	2.14	0.57	3.14
High Calories, 1-0	2.1389	**0.0023**	1	2.14	0.77	3.51
	**Variables excluded from the model and reasons for** **exclusion**					
Arm circumference	Component of Fitscore					
Heart rate	Component of Fitscore					
Vital capacity	Component of Fitscore					
Tricipital skinfold	Used to clean arm circumference from subcutaneous fat					
Forced expiratory volume	Collinearity problems with vital capacity					
Dietary score (3 classes)	Collinearity problems with Calories					

1-0: reported for dichotomic variables, with mean 1 = yes and 0 = no. Delta = amount of variable used to estimate the effect (around standard deviation for continuous variables). Effect = years gained (+) or lost (−) as a function of variable change defined by delta. CI = confidence interval. In **bold** are significant *p* values. (*) Criteria derived from the ECG Minnesota Code [7].

## Data Availability

The data and computing codes are not available for replication because the original data are not publicly available, although the Board of Directors of the study may evaluate specific requests for dedicated analyses.

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
