# Peer review of "Physical Activity, Physical Fitness and Energy Intake Predict All-Cause Mortality and Age at Death in Extinct Cohorts of Middle-Aged Men Followed-Up for 61 Years"

_jcm, 2025, doi:10.3390/jcm14072178_

Round 1

Reviewer 1 Report

Comments and Suggestions for Authors

This is an interesting study with very long follow-up (61 years) that followed 1712 subjects and 3 indexes of physical activity, physical fitness, and energy intake to predict all-cause mortality and age of death in extinct cohort of middle-aged men. They show survival among these grups showing that does that were in class 3 (highest Phyac and Fitscore) had best survival while the opposite was true for calories intake.

There are some notable shortcomings to this study design, despite impressive follow-up, as outlined below.

1. Basaline differences between compared groups of interest are not shown and provided and this can induce serious bias as we do not know if these groups differed systematically in some important confounding variables and unmeasured/undisclosed parameters.

2. There is a lack of information on comorbidities, cancer, and many other important medical history items that could have affected survival profoundly.

3. Authors acknowledge no limitations to their study and this section should be added for sure.

4. During this long time, many of the variables are prone to dynamic change in time, therefore, this bias should be accounted for.

5. According to these limits, conclusions should be toned down.

Author Response

Assigned Editor Miana Zhou

Journal Journal of Clinical Medicine

Manuscript ID jcm-3371768

Title: PHYSICAL ACTIVITY, PHYSICAL FITNESS AND ENERGY INTAKE PREDICT ALL-CAUSE MORTALITY AND AGE AT DEATH IN EXTINCT COHORTS OF MIDDLE-AGED MEN FOLLOWED-UP FOR 61 YEARS

Note from the authors for the three Referees

Some minor changes and corrections have been added beyond those suggested by the Referees. Moreover, in the revised Discussion there has been an exchange in the sequence of some paragraphs.

Following a request of the Editorial Office we made a reduction of our personal bibliographic references (self-citations), from 7 to 5, thus reducing also the total from 28 to 26. As a consequence, the sequence of references has been subverted , with those changed marked in yellow.

Referee 1

Comments and Suggestions for Authors

This is an interesting study with very long follow-up (61 years) that followed 1712 subjects and 3 indexes of physical activity, physical fitness, and energy intake to predict all-cause mortality and age of death in extinct cohort of middle-aged men. They show survival among these grups showing that does that were in class 3 (highest Phyac and Fitscore) had best survival while the opposite was true for calories intake.

There are some notable shortcomings to this study design, despite impressive follow-up, as outlined below.

QUESTION 1. Baseline differences between compared groups of interest are not shown and provided and this can induce serious bias as we do not know if these groups differed systematically in some important confounding variables and unmeasured/undisclosed parameters.

QUESTION 2. There is a lack of information on comorbidities, cancer, and many other important medical history items that could have affected survival profoundly.
ANSWER 1 and 2. The first 2 questions relate to the limited description of other characteristics of the various groups. The fact is that although baseline data were collected 65 years ago, the available ones are relatively large, but dealing with what was available, known and open to be standardized those times (the early stage of field epidemiology in most countries).

To avoid enormous tabulations, the authors decided to provide this description in an indirect way, that is :

-leaving the present analysis as it is

- but adding a multiple regression model with age at death as end-point (considering that the cohort is practically extinct)

-and adding a large number of risk factors, personal characteristics, and morbid conditions partly related to age at death

-including, of course physical activity, fitscore and calories,

-excluding variables included in the firtscore and other that are highly related to calories intake.

QUESTION 3. Authors acknowledge no limitations to their study and this section should be added for sure.

ANSWER 3. In the revised Discussion a chapter has been added on the limitation of the study.

QUESTION 4. During this long time, many of the variables are prone to dynamic change in time, therefore, this bias should be accounted for.

ANSWER 4. The study performed another field examination around mid-way of the follow-up (at year 31) but only a quarter of the original population was surviving and some critical variables for the analysis were not  measured again, thus preventing from studying the effect of risk factor changes. The problem has been considered in the revised Discussion.

QUESTION 5. According to these limits, conclusions should be toned down.

ANSWER 5. In the revised Discussion the final conclusions have been modified consequently.

Reviewer 2 Report

Comments and Suggestions for Authors

This study investigates the long-term impact of physical activity, physical fitness, and energy intake on all-cause mortality and age at death in middle-aged men. The study is based on two residential cohorts from the Italian Rural Areas of the Seven Countries Study (SCS), initiated in 1960, and follows 1,712 men aged 40–59 for 61 years.  

The three primary variables analyzed were: physical activity (Phyac), classified into sedentary, moderate, and vigorous categories based on occupation; physical fitness (Fitscore), calculated using arm circumference, resting heart rate, and vital capacity through Principal Component Analysis (PCA); and energy intake (Calories), estimated from dietary history and categorized into low, intermediate, and high intake. Five additional cardiovascular risk factors—age, smoking, BMI, blood pressure, and cholesterol—were included as covariates.  

The results indicate strong correlations among Phyac, Fitscore, and Calories. Kaplan-Meier survival analysis revealed that higher levels of Phyac, Fitscore, and Calories were independently associated with lower mortality and longer lifespan. Cox proportional hazards models confirmed their protective effects, showing significant negative hazard ratios for all three variables, even after adjusting for confounders. Similarly, multiple linear regression models predicting age at death demonstrated significant positive coefficients, suggesting that greater physical activity, fitness, and energy intake contribute to longevity.  

An unexpected but notable finding was that higher energy intake correlated with increased lifespan, likely due to high energy expenditure from physically demanding occupations. However, the study is limited by the lack of leisure physical activity data, the historical context of rural labor-intensive lifestyles, and the absence of direct fitness testing.  

The study concludes that occupational physical activity, fitness indicators, and energy intake are independent and complementary predictors of longevity. Future research should explore gender differences, modern lifestyle changes, and the role of leisure physical activity in mortality risk.

I believe there are some issues that must be addressed:

In the abstract, the authors should explicitly state the sample size (1,712 participants) and the number of deaths observed (1,708). Additionally, the abstract conclusion requires more precision regarding the relative contributions of each variable to mortality risk.  

Figure 1: The Kaplan-Meier survival curves should display confidence intervals and the number of at-risk participants at different time points to enhance interpretability.  

 Table 4: Hazard ratios (HRs) are reported, but confidence intervals (CIs) are missing for comparison purposes. Providing all CIs ensures statistical robustness.  

 Table 5: The multiple linear regression (MLR) models should include an adjusted R² value to indicate the proportion of variance explained by the predictors.  

Although not addressable in this cohort, the raw fundamentals of the study are extremely shaky. The study vaguely recalls The Framingham study.

Physical activity classification is extremely unreliable (Lines 63–66). The study relies solely on occupational physical activity, which limits generalizability. The authors acknowledge (Lines 256–258) the absence of leisure physical activity data, but its exclusion may lead to an underestimation of overall physical activity effects.  

Energy intake assessment in also extremely unreliable (Lines 75–77). The dietary history method is subject to recall bias. There is also lack of validation versus other dietary assessment methods. 

Survival Analysis is not clearly traceable (Lines 150–156). The study does not discuss potential confounders beyond cardiovascular risk factors. Socioeconomic status, access to healthcare, and genetic predispositions could also influence longevity.  

In consideration of foresight and perseverance shown by the authors addressing some of these limitations in future study’s could greatly benefit and give relevance to contemporary public health research.

Given the study’s focus on aging, integrating frailty scores or gait speed assessments could provide deeper insights into how physical fitness translates into longevity.

Future research should also explore whether similar trends hold for women, as the study is male-only.  

Given the shift from labor-intensive occupations to sedentary lifestyles, future studies should assess the impact of sedentary behavior and modern leisure physical activity patterns on longevity.  

Author Response

Assigned Editor Miana Zhou

Journal Journal of Clinical Medicine

Manuscript ID jcm-3371768

Title: PHYSICAL ACTIVITY, PHYSICAL FITNESS AND ENERGY INTAKE PREDICT ALL-CAUSE MORTALITY AND AGE AT DEATH IN EXTINCT COHORTS OF MIDDLE-AGED MEN FOLLOWED-UP FOR 61 YEARS

Note from the authors for the three Referees

Some minor changes and corrections have been added beyond those suggested by the Referees. Moreover, in the revised Discussion there has been an exchange in the sequence of some paragraphs.

Following a request of the Editorial Office we made a reduction of our personal bibliographic references (self-citations), from 7 to 5, thus reducing also the total from 28 to 26. As a consequence, the sequence of references has been subverted , with those changed marked in yellow.

Referee 2

Comments and Suggestions for Authors

This study investigates the long-term impact of physical activity, physical fitness, and energy intake on all-cause mortality and age at death in middle-aged men. The study is based on two residential cohorts from the Italian Rural Areas of the Seven Countries Study (SCS), initiated in 1960, and follows 1,712 men aged 40–59 for 61 years.  

The three primary variables analyzed were: physical activity (Phyac), classified into sedentary, moderate, and vigorous categories based on occupation; physical fitness (Fitscore), calculated using arm circumference, resting heart rate, and vital capacity through Principal Component Analysis (PCA); and energy intake (Calories), estimated from dietary history and categorized into low, intermediate, and high intake. Five additional cardiovascular risk factors—age, smoking, BMI, blood pressure, and cholesterol—were included as covariates.  

The results indicate strong correlations among Phyac, Fitscore, and Calories. Kaplan-Meier survival analysis revealed that higher levels of Phyac, Fitscore, and Calories were independently associated with lower mortality and longer lifespan. Cox proportional hazards models confirmed their protective effects, showing significant negative hazard ratios for all three variables, even after adjusting for confounders. Similarly, multiple linear regression models predicting age at death demonstrated significant positive coefficients, suggesting that greater physical activity, fitness, and energy intake contribute to longevity.  

An unexpected but notable finding was that higher energy intake correlated with increased lifespan, likely due to high energy expenditure from physically demanding occupations. However, the study is limited by the lack of leisure physical activity data, the historical context of rural labor-intensive lifestyles, and the absence of direct fitness testing.  

The study concludes that occupational physical activity, fitness indicators, and energy intake are independent and complementary predictors of longevity. Future research should explore gender differences, modern lifestyle changes, and the role of leisure physical activity in mortality risk.

I believe there are some issues that must be addressed:

QUESTION 1. In the abstract, the authors should explicitly state the sample size (1,712 participants) and the number of deaths observed (1,708). Additionally, the abstract conclusion requires more precision regarding the relative contributions of each variable to mortality risk.

ANSWER 1. The number of the sample size and of deaths have been added in the Abstract. More information on the role of the three indexes has been added in the revised Abstract.

QUESTION 2. Figure 1: The Kaplan-Meier survival curves should display confidence intervals and the number of at-risk participants at different time points to enhance interpretability.

ANSWER 2. Considering the size of the Figure in a printed page, the addition of confidence limits to 3 lines, means a mess of 9 lines largely overlapping and therefore we excluded this option. However, the numbers of survivors  at different times have been added.

QUESTION 3. Table 4. Hazard ratios (HRs) are reported, but confidence intervals (CIs) are missing for comparison purposes. Providing all CIs ensures statistical robustness.

ANSWER 3. Unfortunately, the confidence intervals were already reported and therefore it is not clear what is the meaning of this request.

QUESTION 4. Table 5: The multiple linear regression (MLR) models should include an adjusted R² value to indicate the proportion of variance explained by the predictors.  

ANSWER 4. R2 have been added with the outcome of MLR models.

QUESTION 5. Although not addressable in this cohort, the raw fundamentals of the study are extremely shaky. The study vaguely recalls The Framingham study.

Physical activity classification is extremely unreliable (Lines 63–66). The study relies solely on occupational physical activity, which limits generalizability. The authors acknowledge (Lines 256–258) the absence of leisure physical activity data, but its exclusion may lead to an underestimation of overall physical activity effects.  

Energy intake assessment in also extremely unreliable (Lines 75–77). The dietary history method is subject to recall bias. There is also lack of validation versus other dietary assessment methods. 

Survival Analysis is not clearly traceable (Lines 150–156). The study does not discuss potential confounders beyond cardiovascular risk factors. Socioeconomic status, access to healthcare, and genetic predispositions could also influence longevity.  

In consideration of foresight and perseverance shown by the authors addressing some of these limitations in future study’s could greatly benefit and give relevance to contemporary public health research.

ANSWER 5. It should be taken into account that baseline measurements were made 65 years ago when techniques and procedures were definitely poor. Leisure physical activity did not exist as it is the case nowadays where the situation is entirely different.

The technical procedure for the classification of physical activity was extremely rough, but those times (i.e. 65 years ago) valuable tools for the purpose were not available, although even more recently there are studies using a classification of physical activity based on a single question or self-reported information. A partial validation based on ergonometric procedures has already been quoted.

The dietary history used in the dietary survey is considered not reliable, but those time it was among the best available tools. If so, we should forget the majority of dietary studies of the last century, beyond considering that many recent investigations are based on questionnaires that look for the frequency of eaten food groups.

It is not clear why the survival analysis is not clearly traceable considering that a full paragraph and 3 figures describe this issue.

Notes on the above problems have been added in the revised Discussion.

The complaint about the limited number of other possible confounders was risen also by Reviewer N.1 and therefore we present here the same solution. To avoid enormous tabulations, the authors decided to provide this description in an indirect way, that is :

-leaving the present analysis as it is

-but adding a multiple regression model with age at death as end-point (considering that the cohort is practically extinct)

-and adding a large number of risk factors, personal characteristics, and morbid conditions  partly related to age at death

-including, of course physical activity, fitscore and calories,

-excluding variables included in the firtscore and other that are highly related to calories intake.

Moreover, in the revised Discussion there are several paragraphs discussion the limitations of the analysis but also the perspective for the future.

QUESTION 6

Given the study’s focus on aging, integrating frailty scores or gait speed assessments could provide deeper insights into how physical fitness translates into longevity.

Future research should also explore whether similar trends hold for women, as the study is male-only.  

Given the shift from labor-intensive occupations to sedentary lifestyles, future studies should assess the impact of sedentary behavior and modern leisure physical activity patterns on longevity.  

ANSWER 6. Examinations dedicated to frailty and similar problems of the elderly were not performed by our research group but this analysis in focused on characteristics allowing a long expectancy of life and not to describe the characteristics of those who reached old ages when this goal was obtained.

Reviewer 3 Report

Comments and Suggestions for Authors

In this interesting paper, the authors evaluated the effect of working physical activity, physical fitness and energy intake on all-cause mortality and age at death in cohorts of middle-aged men followed-up for 61 years.

Coefficients of Phyac, Fitscore and Calories were all positive and significant with beneficial effects of their high levels.

The authors demonstrated the protective role of working vigorous physical activity versus all-cause mortality and age at death. They observed a favourable relationship of energy intake with mortality and age at death.

The authors used arm circumference, heart rate and vital capacity as objective indicators of muscolar, cardio-circulatory and respiratory fitness. 

The strength of the manuscript is the follow-up duration (61 years).

The authors adjusted Phyac, Fitscore and Calories for age, smoking, BMI, SBP and serum cholesterol. 

Overall, the mansucript is well written, the statistics is adequate, tables and figures are well presented, the references are appropriate and the conclusions clearly summarize the main findings of the study.

Among the anthropometrics, the authors considered BMI but they did not consider the chest wall conformation.

What about the potential role of chest shape conformation versus the endpoint all-cause mortality in the general population?

At the end of the Discussion section, the authors could discuss the opposite role of a concave-shaped chest wall conformation and/or narrow antero-posterior thoracic diameter (commonly detected in individuals with mitral valve prolapse and low prevalence of obstructive coronary artery disease) (PMID: 34485034) and, conversely, the role of a more spheroidal thoracic shape conformation (commonly observed among elderly males with asymptomatic atrial fibrillation and/or respiratory diseases) (PMID: 25534423) versus the occurrence of adverse cardiovascular events. 

It is important to consider not only the BMI but also the chest wall conformation, that may help the clinicians to identify different phenotypes of individuals, with different risk of adverse cardiovascular events.

Comments on the Quality of English Language

The quality of English language is good.

Author Response

Assigned Editor Miana Zhou

Journal Journal of Clinical Medicine

Manuscript ID jcm-3371768

Title: PHYSICAL ACTIVITY, PHYSICAL FITNESS AND ENERGY INTAKE PREDICT ALL-CAUSE MORTALITY AND AGE AT DEATH IN EXTINCT COHORTS OF MIDDLE-AGED MEN FOLLOWED-UP FOR 61 YEARS

Note from the authors for the three Referees

Some minor changes and corrections have been added beyond those suggested by the Referees. Moreover, in the revised Discussion there has been an exchange in the sequence of some paragraphs.

Following a request of the Editorial Office we made a reduction of our personal bibliographic references (self-citations), from 7 to 5, thus reducing also the total from 28 to 26. As a consequence, the sequence of references has been subverted , with those changed marked in yellow.

Referee 3

1st PARTIn this interesting paper, the authors evaluated the effect of working physical activity, physical fitness and energy intake on all-cause mortality and age at death in cohorts of middle-aged men followed-up for 61 years.

Coefficients of Phyac, Fitscore and Calories were all positive and significant with beneficial effects of their high levels.

The authors demonstrated the protective role of working vigorous physical activity versus all-cause mortality and age at death. They observed a favourable relationship of energy intake with mortality and age at death.

The authors used arm circumference, heart rate and vital capacity as objective indicators of muscolar, cardio-circulatory and respiratory fitness. 

The strength of the manuscript is the follow-up duration (61 years).

The authors adjusted Phyac, Fitscore and Calories for age, smoking, BMI, SBP and serum cholesterol. 

Overall, the manuscript is well written, the statistics is adequate, tables and figures are well presented, the references are appropriate and the conclusions clearly summarize the main findings of the study.

ANSWER 1st PART.

The authors thank for the appreciation.

2nd PART. Among the anthropometrics, the authors considered BMI but they did not consider the chest wall conformation.

What about the potential role of chest shape conformation versus the endpoint all-cause mortality in the general population?

At the end of the Discussion section, the authors could discuss the opposite role of a concave-shaped chest wall conformation and/or narrow antero-posterior thoracic diameter (commonly detected in individuals with mitral valve prolapse and low prevalence of obstructive coronary artery disease) (PMID: 34485034) and, conversely, the role of a more spheroidal thoracic shape conformation (commonly observed among elderly males with asymptomatic atrial fibrillation and/or respiratory diseases) (PMID: 25534423) versus the occurrence of adverse cardiovascular events. 

It is important to consider not only the BMI but also the chest wall conformation, that may help the clinicians to identify different phenotypes of individuals, with different risk of adverse cardiovascular events.

ANSWER  2nd PART. The problem of the chest wall conformation was not considered nor know by the investigators to started the study 65 years ago. The authors believe that a discussion on this issue may take too far away from the basic problem of physical activity, physical fitness and energy intake.

Round 2

Reviewer 1 Report

Comments and Suggestions for Authors

Thank you for addressing all my concerns. No further questions.

Author Response

We appreciate your comments and conclusions and thank for your positive evaluation.

Reviewer 2 Report

Comments and Suggestions for Authors

The authors have made substantial improvements, particularly in reporting statistical robustness and acknowledging study limitations by addressing my main concerns. 

The abstract has been improved accordingly. The conclusion in the abstract still lacks precise quantification of the relative contributions of each variable to mortality risk. It states that all three factors (physical activity, fitness, and energy intake) contribute to lower mortality but does not specify their comparative strengths.

The Kaplan-Meier survival curves are included in Figures 1, 2, and 3. However, confidence intervals and the number of participants at risk at different time points are not reported in the figures.

Table 4 and Table 5 have been adjusted. The authors acknowledge the limitation related to physical activity and dietary history method. While the study includes additional risk factors (e.g., socioeconomic status and major diseases in Table 6), factors like healthcare access and genetic predispositions are not explicitly accounted for.

The discussion section acknowledges the need for future studies to include women and explore the impact of modern sedentary lifestyles.

Author Response

See the included file.
